# Serum miRNAs associated with tumor-promoting cytokines in non-small cell lung cancer

Pichitpon Chaniad[1], Keson Trakunran[1], Sarayut Lucien Geater[2],
Warangkana Keeratichananont[2], Paramee Thongsuksai[3], Pritsana Raungrut[1] *

**1** Department of Biomedical Science, Faculty of Medicine, Prince of Songkla University, Songkhla, Thailand,
**2** Division of Respiratory and Respiratory Critical Care Medicine, Faculty of Medicine, Prince of Songkla University, Songkhla, Thailand, **3** Department of Pathology Department, Faculty of Medicine, Prince of Songkla University, Songkhla, Thailand

* rpritsan@medicine.psu.ac.th

University, TURKEY

**Data Availability Statement:** All relevant data are
within the manuscript and its Supporting
information files.

## Abstract

Tumor-promoting cytokines are a cause of tumor progression; therefore, identifying key regulatory microRNAs (miRNAs) for controlling their production is important. The aim of this study is to identify promising miRNAs associated with tumor-promoting cytokines in non-small cell lung cancer (NSCLC). We identified circulating miRNAs from 16 published miRNA profiles. The selected miRNAs were validated in the serum of 32 NSCLC patients and compared with 33 patients with other lung diseases and 23 healthy persons using quantitative real-time PCR. The cytokine concentration was investigated using the enzyme-linked immunoassay in the same sample set, with clinical validation of the miRNAs. The correlation between miRNA expression and cytokine concentration was evaluated by Spearman's rank correlation. For consistent direction, one up-regulated miRNA (miR-145) was found in four studies, and seven miRNAs were reported in three studies. One miRNA (miR-20a) and four miRNAs (miR-25-3p, miR-223, let-7f, and miR-20b) were reported in six and five studies. However, their expression was inconsistent. In the clinical validation, serum miR-145 was significantly down-regulated, whereas serum miR-20a was significantly up-regulated in NSCLC, compared with controls. Regarding serum cytokine, all cytokines [vascular endothelial growth factor (VEGF), interleukin-6 (IL-6), and transforming growth factor β (TGF-β)], except tumor necrosis factor-α (TNF-α), had a higher level in NSCLC patients than controls. In addition, we found a moderate correlation between the TGF-β concentration and miR-20a ($r = -0.537$, $p = 0.002$) and miR-223 ($r = 0.428$, $p = 0.015$) and a weak correlation between the VEGF concentration with miR-20a ($r = 0.376$, $p = 0.037$) and miR-223 ($r = -0.355$, $p = 0.046$). MiR-145 and miR-20a are potential biomarkers for NSCLC. In addition, the regulation of tumor-promoting cytokine, through miR-20a and miR-223, might be a new therapeutic approach for lung cancer.

**Funding:** This study is funded by a grant from the Prince of Songkla University [grant number MED590692S], the Research Center for Cancer Control in Thailand [grant number MEDRC59036], and the Faculty of Medicine, Prince of Songkla University [grant number REC60350042]. The funders had no role in study design, data collection and analysis, decision to publish, or preparation of the manuscript.

**Competing interests:** The authors have declared that no competing interests exist.

## Introduction

Lung cancer is the most common and leading cause of cancer death worldwide, accounting for 11.6% of the total cases and 18.4% of the total cancer deaths in 2018, respectively [1]. Approximately 84% of lung cancer is non-small cell lung cancer (NSCLC), which is usually diagnosed in the advanced stage (30%–79% of all NSCLC cases) and accompanied by lymph node and distant metastasis [2, 3]. Tumor progression is supported by tumor-promoting cytokines, in which high serum levels of several cytokines, such as vascular endothelial growth factor (VEGF), interleukin-6 (IL-6), transforming growth factor β (TGF-β), and tumor necrosis factor-α (TNF-α), are correlated with advanced stages of lung cancer [4, 5]. Currently, there is accumulating evidence showing that these cytokines are regulated by microRNAs (miRNAs). Therefore, the identification of miRNAs associated with cytokine production is desired.

MiRNAs are short noncoding RNAs, 20–22 nucleotides in length. They suppress gene expression at the post-transcriptional level by binding to the 3'-untranslated regions of mRNAs [6]. Numerous studies have shown that miRNAs play a part in cell differentiation, proliferation, apoptosis, and cytokine production [6, 7]. In addition, the remarkable stability of serum miRNAs under various conditions was found in our previous report [8]. Several studies have revealed that the expression levels display oscillatory changes in response to TNF-α in macrophage cells [9], and VEGF and IL-6 in bone marrow mesenchymal stromal cells [10]. Moreover, changes in miRNA expression have been demonstrated to be involved with cytokine signaling pathways in several types of cancers, including colorectal [11], breast [12], and lung [13] cancers. Although there has been accumulating evidence to support the regulation of cytokines by miRNAs, studies conducted to determine the association of miRNAs and tumor-promoting cytokines in clinical samples in lung cancer are limited.

In this study, we identified differentially expressed (DE) miRNAs through the systematic review of previously published studies on miRNA profiling in NSCLC. A vote-counting procedure and bioinformatics analysis were used to obtain miRNAs associated with cytokine signaling pathways. We then validated the selected miRNAs using quantitative real-time PCR in serum samples of NSCLC compared with controls, which included patients with other lung diseases (OL) and healthy persons (HP). We also determined the serum concentration of the common tumor-promoting cytokines (IL-6, VEGF, TGF-β, and TNF-α) using the enzyme-linked immunoassay (ELISA) in samples from the same patients with miRNA validation. Finally, we investigated the association of miRNA expression and cytokine concentration in the serum of NSCLC patients.

## Materials and methods

### Literature search

The literature on miRNA profiling of lung cancer, published between 2006 and 2016, was retrieved from "PubMed" and "Scopus" databases. The search terms "miRNA or miR or microRNA," "lung cancer," and "profiling" were used.

### Study selection criteria

Eligible studies were required to meet the following inclusion criteria: 1) miRNA expression profiling of patients with NSCLC; 2) conducted on blood, serum, or plasma; 3) used samples from patients with OL or HP for comparison; 4) were full-text articles in English. The exclusion criteria were: 1) patients received any previous treatment; 2) studies that did not report p-values, or reported a false discovery rate (FDR); 3) studies were review articles; 4) studies that did not report sample size.

## Data extraction

Lists of DE miRNAs were extracted from each selected article. Related information was also retrieved, including authors, year of publication, country, specimen type, histological subtypes, number of samples (both cases and controls), stage of cancer, array platforms, cut-off criteria (*p*-value or FDR), and numbers of DE miRNAs. All articles were independently assessed by two researchers (Chaniad P and Raungrut P).

## Ranking

The vote-counting strategy reported in Griffith's and Chan's studies [14, 15] was used. The DE miRNAs were ranked for importance in the following order: 1) the number of articles that consistently reported as differentially expressed; 2) direction of change of DE miRNA, and 3) the total number of samples. Moreover, we considered the number of target genes supported by reliable and experimental evidence from the miRTarBase 7.0 [16].

## Bioinformatics analysis

The miRTarbase 7.0 containing experimentally validated miRNA-target interactions was used to identify potential gene targets [16]. Enrichment analyses of Gene Ontology (GO) terms and Kyoto Encyclopedia of Genes and Genomes (KEGG) pathways were analyzed by DIANA-miRPath v3.0. [17]. The integrated genes regulatory network was generated by Cytoscape 3.5.1 [18]. Cytokine-related genes were retrieved from the Mouse Genome Informatics Web Site [19].

## Study subjects and sample collection

This study was approved by the Ethics Committee on Human Research, Faculty of Medicine, Prince of Songkla University (REC 59-210-04-2 and REC 59-211-04-2). All subjects, including 32 NSCLC patients, 33 OL patients, and 23 HP, were recruited from Songklanagarind Hospital, Songkhla, Thailand, from May 2016 to April 2017. NSCLC was diagnosed according to 2015 WHO classification of lung and pleural tumors [20]. The staging of cancer was based on the Tumor Node Metastasis cancer staging system of the AJCC Cancer Staging Manual (7[th] Edition) [21]. No patients received any treatment before sample collection.

For the control groups, subjects who were age- and gender-matched with the NSCLC group were included. The OL group were patients who had symptoms similar to lung cancer, such as tuberculosis, bronchiectasis, interstitial lung disease, pneumonia, and chronic obstructive pulmonary disease. The HP group was recruited from people who had an annual checkup for two consecutive years. Written informed consent was obtained from all subjects. Peripheral blood (5 ml) was collected in a clotting tube (Greiner Bio-One, Kremsmünster, Austria). Serum was isolated by centrifugation and prepared for miRNA isolation, as previously described [8].

## Quantitative real-time PCR (qRT-PCR)

The total RNA from serum was extracted using Trizol® LS (Invitrogen, California, USA) according to the manufacturer's protocol. The quantity and quality of the total RNA were measured using a NanoDrop® ND-1000 UV-Vis spectrophotometer (Thermo Scientific, Massachusetts, USA) at an optical density ratio of A260/280 nm and A260/230 nm. The total RNA (50 ng) was reverse transcribed into complementary DNA (cDNA) by the miScript II RT kit (Qiagen, Hilden, Germany). Reverse transcription was performed by Thermal cycler (Bio-Rad, California, USA) at 37 ˚C for 60 mins, and 95 ˚C for 5 mins. The cDNA was diluted by

adding 200 μl of RNase-free water. Subsequently, miRNA amplification was performed using a Bio-Rad CFX96 qPCR system (Bio-Rad, California, USA) with the miScript SYBR$^{®}$ green PCR kit (Qiagen, Hilden, Germany) according to the manufacturer's protocol. Primers for miR-145-5p, miR-206-5p, miR-20a-5p, miR-223-5p, miR-25-3p, let-7f-5p, miR-20b-5p, and U6 small nuclear RNA (RNU6), were used as internal controls, and obtained from Qiagen (Qiagen, Hilden, Germany). Data were presented as the cycle threshold (Ct) value, which was determined using the default threshold settings. The difference between the Ct value (ΔCt) of the miRNA of interest and RNU6 was calculated, and the relative expression was presented using the $2^{-\Delta CT}$ method [22].

### Enzyme-linked immunoassay

Concentrations of VEGF, IL-6, TGF-β, and TNF-α were measured quantitatively using 100 μl of serum with a sandwich human ELISA kit (Prepotech, Rehovot, Israel) according to the manufacturer's instructions. Absorbance was read at 405 nm with a reference wavelength of 650 nm by a microplate reader (Molecular Devices, California, USA). Samples were assayed in duplicate, and concentrations of each cytokine were calculated by the constructed standard curve.

### Statistical analysis

Data distribution of relative miRNA expression and cytokine concentration was assessed by the Shapiro–Wilk normality test. Outliers of the data were detected using the Grubbs' test. Differences in the data between the groups of subjects were analyzed by the Student's *t*-test if the data had a normal distribution, and by the Wilcoxon–Mann–Whitney test in cases of non-normal distribution. The correlation between miRNA expression and cytokine concentration was analyzed by the Spearman's rank correlation. A *p*-value of less than 0.05 was considered statistically significant. All plots and statistical analyses were performed using the R-statistical software version 3.4.4.

## Results

### Included articles

Fig 1 depicts the number of selected articles from each selection process. A total of 725 studies were identified using our search terms. There were 536 remaining articles, after removing 189 duplicates. After screening the titles and abstracts, 47 eligible articles were included. According to the exclusion criteria, only 16 articles were finally included in our analysis (Fig 1). All characteristics of these articles are shown in Table 1.

### Differentially expressed miRNAs

In total, 229 DE miRNAs were obtained from the 16 miRNA profiles studied, of which 181 and 48 miRNAs were reported with consistent and inconsistent direction (both up- and down-regulation), respectively. Among the 181 consistent miRNAs, 97 miRNAs were up-regulated, and 84 miRNAs were down-regulated.

Only miR-145 was reported to be consistently up-regulated in four studies. Six miRNAs (miR-320, miR-151a, miR-16, miR-200b-3p, miR-205, and miR-574) and one miRNA (miR-1285-3p) were most consistently reported to be up-regulated and down-regulated in three studies, respectively (Table 2). For inconsistent direction, miR-20a was reported in six studies (4 up-/2 down-regulated expression), and miRNAs (miR-25-3p, miR-223, let-7f, and miR-20b) were reported in five studies.

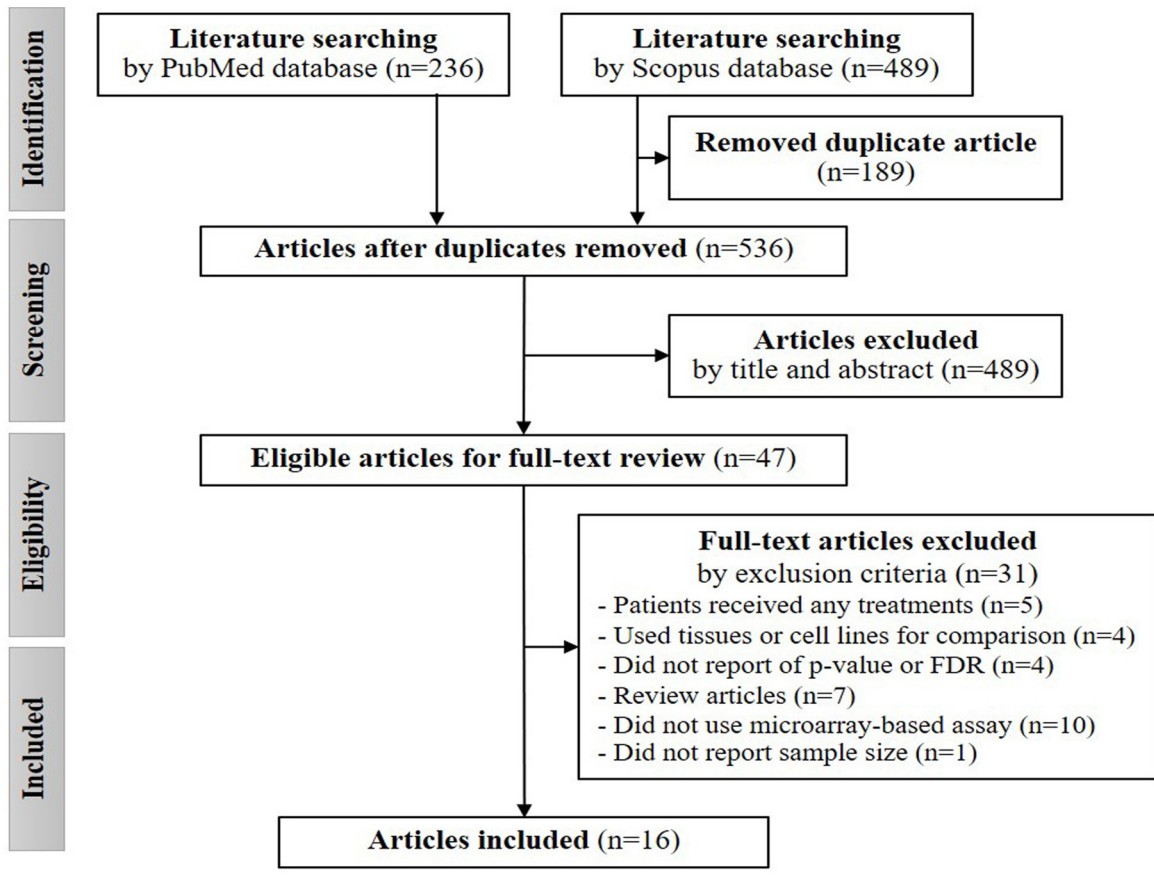

**Fig 1. The study selection process.**

List of potential total genes and targeted genes that are supported by strong and experimental evidences was presented in S1 Table for the consistently up-regulated miRNA, S2 Table for the consistently down-regulated miRNA, and S3 Table for the inconsistency reported miRNA.

## Functional enrichment analysis and gene interaction network

We performed a functional enrichment analysis to find the molecular networks and their target genes of seven DE miRNAs, including miR-145, miR-206, miR-20a, miR-223, miR-25-3p, let-7f, and miR-20b. The most significantly enriched GO terms on molecular functions, cellular components, and biological functions are iron binding (1662 genes), organelles (2973 genes), and cellular nitrogen metabolic processes (1499 genes) (Fig 2A). Also, the KEGG pathway showed that target genes were enriched in pathways for cancer (155 genes), PI3K-Akt signaling pathways (110 genes), proteoglycans in cancer (97 genes), MAPK signaling pathways (88 genes), and protein processing in the endoplasmic reticulum (Fig 2B). The complexity of the miRNA-mRNA interaction network is shown in Fig 2C. The results showed that a single miRNA regulates numerous genes. The highest number of target genes were identified for miR-145 (143 genes), followed by miR-20a (71 genes). In addition, these miRNAs work synergistically. We also revealed that our selected miRNAs regulated cytokine-associated genes, such as VEGFA, IGF1R, TGFBR1, and HIF1A.

**Table 1. Characteristics of included studies.**

| Study | Year | Country | Specimen type | Histological subtype | No. of sample (case/control) | Stage | Platform | Cut-off criteria | DE miRNA Up | Down | Total |
|---|---|---|---|---|---|---|---|---|---|---|---|
| Fan et al. [23] | 2016 | China | Serum | NSCLC | 152 (94/HP 58) | II, IIA-IIIB | Fluorescent coding liquid beads array (Shanghai Tellgen Life Science) | $p < 0.01$ | 6 | 1 | 7 |
| Gao et al. [24] | 2016 | China | Plasma | SCC | 10 (5/HP 5) | I | Taqman low-density array (Applied Biosystems) | $p < 0.01$ | 12 | 6 | 18 |
| Rita et al. [25] | 2016 | Norway | Serum | ADC | 70 (38/HP 16, COPD 16) | I-IV | Taqman low-density array (Applied Biosystems) | $FDR < 0.01$ | 10 | 27 | 37 |
| Leidinger et al.[26] | 2015 | Germany | Blood | NSCLC | 94 (74/HP 20) | I-IV | 96.96 Dynamic array (Fluidigm) | $p < 0.05$ | 15 | 16 | 31 |
| | | | | | 100 (74/COPD 26) | | | | 10 | 21 | 31 |
| Nadal et al. [27] | 2015 | American | Serum | NSCLC | 92 (70/HP 22) | I-III | mirVana bioarrays v2.0 (Ambion) | $p \leq 0.001$ | 60 | 31 | 91 |
| Wozniak et al.[28] | 2015 | Russia | Plasma | NSCLC | 200 (100/HP 100) | IA-IIIA | TaqMan Human MicroRNA Array A + B Card Set v3.0 (Applied Biosystems) | $p < 0.05$ | 33 | 28 | 61 |
| Geng et al. [29] | 2014 | China | Plasma | NSCLC | 50 (25/HP 25) | I-II | microarray (Qiagen) | $p < 0.05$ | 12 | 0 | 12 |
| Rani et al. [30] | 2013 | Ireland | Serum | ADC | 80 (40/HP 40) | I-IV | Taqman low-density array (Applied Biosystems) | $p < 0.05$ | 6 | 2 | 8 |
| Heegaard et al.[31] | 2012 | Denmark | Serum | NSCLC | 440 (220/HP 220) | I + IA-IIB | 96.96 Dynamic array (Fluidigm) | $p < 0.05$ | 1 | 7 | 8 |
| Patnaik et al. [32] | 2012 | American | Blood | ADC | 45 (22/HP 23) | IA-IIIB | miRCURY locked nucleic acid microarrays (Exiqon) | $FDR < 0.01$ | 12 | 12 | 24 |
| Roth et al. [33] | 2012 | Germany | Serum | NSCLC | 32 (21/HP 11) | I-IV | Microfluid biochips (Febit Biomed GmbH) | $p < 0.05$ | 18 | 12 | 30 |
| Chen et al. [34] | 2012 | China | Serum | NSCLC | 310 (200/HP 110) | I-IV | Taqman probe-based qRT-PCR assay (Applied Biosystems) | $p < 0.05$ | 10 | 0 | 10 |
| Foss et al. [35] | 2011 | America | Serum | NSCLC | 22 (11/HP 11) | I-II | GenoExplorer microRNA Expression System (GenoSensor Corperation) | $p < 0.05$ | 8 | 0 | 8 |
| Silva et al. [36] | 2011 | Spain | Plasma | NSCLC | 48 (28/HP 20) | I-IV | Taqman low-density array (Applied Biosystems) | $p < 0.05$ | 0 | 10 | 10 |
| Wang et al. [37] | 2011 | China | Serum | NSCLC | 10 (5/HP 5) | I-III | Microarray chip (LC sciences) | $FDR < 0.05$ | 11 | 8 | 19 |
| Keller et al. [38] | 2009 | Germany | Blood | NSCLC | 36 (17/HP 19) | I-IV | Geniom Biochip miRNA homo sapiens (Febit Biomed GmbH) | $p < 0.05$ | 13 | 14 | 27 |

NSCLC, non-small cell lung cancer; ADC, adenocarcinoma; SCC, squamous cell carcinoma; HP, healthy person; COPD, chronic obstructive pulmonary disease; FDR, false discovery rate; DE miRNA, differentially expressed miRNA.

## Validation of selected differentially expressed miRNAs

The expression level of seven miRNAs, including miR-145, miR-206, miR-20a, miR-223, miR-25-3p, let-7f, and miR-20b, was validated the serum of 32 patients with NSCLC, 33 patients with OL, and 23 HP using qRT-PCR analysis. NSCLC patients had a mean age of 59.9 years. Eighteen cases were male, and 14 were female. Histological subtype was 84.4% (27/32 of cases) for adenocarcinoma (ADC) and 15.6% (5/32 of cases) for squamous cell carcinoma (SCC). The majority of patients exhibited stage IV lung cancer (90.6%, 29/32 of cases), whereas one and two cases were stage II and I lung cancer, respectively. The OL group included 22 patients with chronic obstructive pulmonary disease, 6 with bronchiectasis, 3 with tuberculosis, 1 with pneumonia, and 1 with idiopathic pulmonary fibrosis.

**Table 2. Differentially expressed miRNAs according to the ranking criteria.**

| miRNA | Mature sequence | Studies | Total sample (Case/Control) | Reference |
|---|---|---|---|---|
| **Consistently up-regulated miRNA** | | | | |
| miR-145 | 16\|GUCCAGUUUUCCCAGGAAUCCCU\|38 | 4 | 572(369/203) | [26, 27, 29, 34] |
| miR-320a | 48\|AAAAGCUGGGUUGAGAGGGCGA\|69 | 3 | 442 (295/147) | [27, 29, 34] |
| miR-151a-3p | 47\|CUAGACUGAAGCUCCUUGAGG\|67 | 3 | 302 (175/127) | [24, 27, 28] |
| miR-16 | 10\|UAGCAGCACGUAAAUAUUGGCG\|31 | 3 | 254 (169/85) | [23, 27, 37] |
| miR-200b-3p | 57\|UAAUACUGCCUGGUAAUGAUGA\|78 | 3 | 172 (113/59) | [25, 27, 37] |
| miR-205 | 34\|UCCUUCAUUCCACCGGAGUCUG\|55 | 3 | 90 (48/42) | [24, 25, 37] |
| miR-574 | 25\|UGAGUGUGUGUGUGUGAGUGUGU\|47 | 3 | 90 (49/41) | [33, 35, 38] |
| miR-125b | 15\|UCCCUGAGACCCUAACUUGUGA\|36 | 2 | 162 (108/54) | [25, 27] |
| miR-186 | 15\|CAAAGAAUUCUCCUUUUGGGCU\|36 | 2 | 162 (108/54) | [25, 27] |
| miR-18a | 47\|ACUGCCCUAAGUGCUCCUUCUGG\|69 | 2 | 156 (91/65) | [26, 38] |
| **Consistently down-regulated miRNA** | | | | |
| miR-1285-3p | 51\|UCUGGGCAACAAAGUGAGACCU\|72 | 3 | 280 (143/137) | [24, 25, 28] |
| miR-1243 | 5\|AACUGGAUCAAUUAUAGGAGUG\|26 | 2 | 292 (170/122) | [27, 28] |
| miR-661 | 51\|UGCCUGGGUCUCUGGCCUGCGCGU\|74 | 2 | 292 (170/122) | [27, 28] |
| miR-708 | 11\|AAGGAGCUUACAAUCUAGCUGGG\|33 | 2 | 292 (170/122) | [27, 28] |
| miR-572 | 61\|GUCCGCUCGGCGGUGGCCCA\|80 | 2 | 140 (98/42) | [27, 36] |
| miR-206 | 53\|UGGAAUGUAAGGAAGUGUGUGG\|74 | 2 | 102 (75/27) | [27, 37] |
| let-7d | 8\|AGAGGUAGUAGGUUGCAUAGUU\|29 | 2 | 84 (45/39) | [36, 38] |
| miR-15a | 14\|UAGCAGCACAUAAUGGUUUGUG\|35 | 2 | 81 (39/42) | [32, 38] |
| **Inconsistently reported miRNA** | | | | |
| miR-20a | 8\|UAAAGUGCUUAUAGUGCAGGUAG\|30 | 4 (Up) | 594 (389/205) | [23, 27, 29, 34] |
| | | 2 (Down) | 168 (97/71) | [26, 32] |
| miR-25-3p | 52\|CAUUGCACUUGUCUCGGUCUGA\|73 | 4 (Up) | 497 (312/176) | [27, 29, 34, 38] |
| | | 1 (Down) | 200 (100/100) | [28] |
| miR-223 | 26\|CGUGUAUUUGACAAGCUGAGUU\|47 | 4 (Up) | 652 (395/257) | [27, 28, 29, 38] |
| | | 1 (Down) | 48 (28/20) | [36] |
| let-7f | 7\|UGAGGUAGUAGAUUGUAUAGUU\|28 | 1 (Up) | 200 (100/100) | [28] |
| | | 4 (Down) | 198 (105/93) | [25, 32, 36, 38] |
| miR-20b | 6\|CAAAGUGCUCAUAGUGCAGGUAG\|28 | 1 (Up) | 92 (70/22) | [27] |
| | | 4 (Down) | 246 (141/105) | [26, 32, 36, 38] |
| let-7a | 57\|CUAUACAAUCUACUGUCUUUC\|77 | 1 (Up) | 21 (21/11) | [33] |
| | | 3 (Down) | 546 (275/ 271) | [25, 31, 38] |
| miR-17 | 14\|CAAAGUGCUUACAGUGCAGGUAG\|36 | 1 (Up) | 92 (70/22) | [27] |
| | | 3 (Down) | 650 (316/289) | [26, 31, 32] |

miR-145 was significantly down-regulated in NSCLC patients compared with OL patients ($p$ = 0.003), HP ($p$ = 0.038), and all controls ($p$ = 0.004) (Fig 3). miR-20a was significantly up-regulated in NSCLC patients compared with OL patients ($p$ = 0.045) and all controls ($p$ = 0.039), but not with those in the HP groups. However, no significant difference was observed for other miRNAs.

## Serum cytokine concentration in NSCLC patients and controls

Concentrations of tumor-promoting cytokines, including IL-6, VEGF, TGF-β, and TNF-α were determined in the serum of NSCLC patients and controls using ELISA. The results revealed that NSCLC patients had significantly higher VEGF concentrations than those of OL

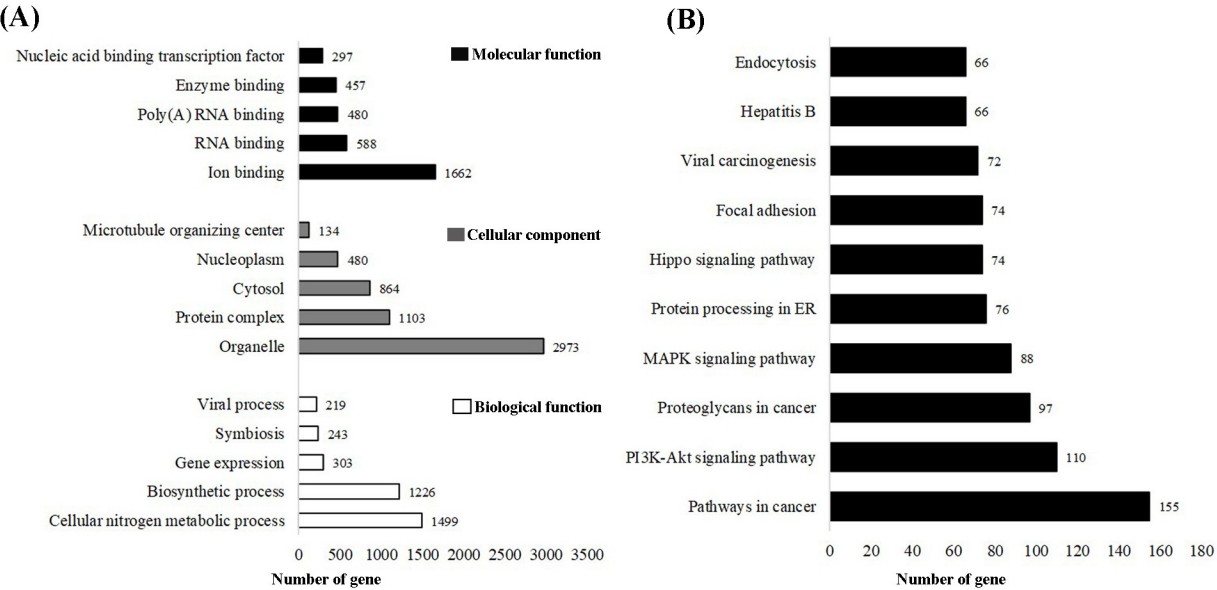

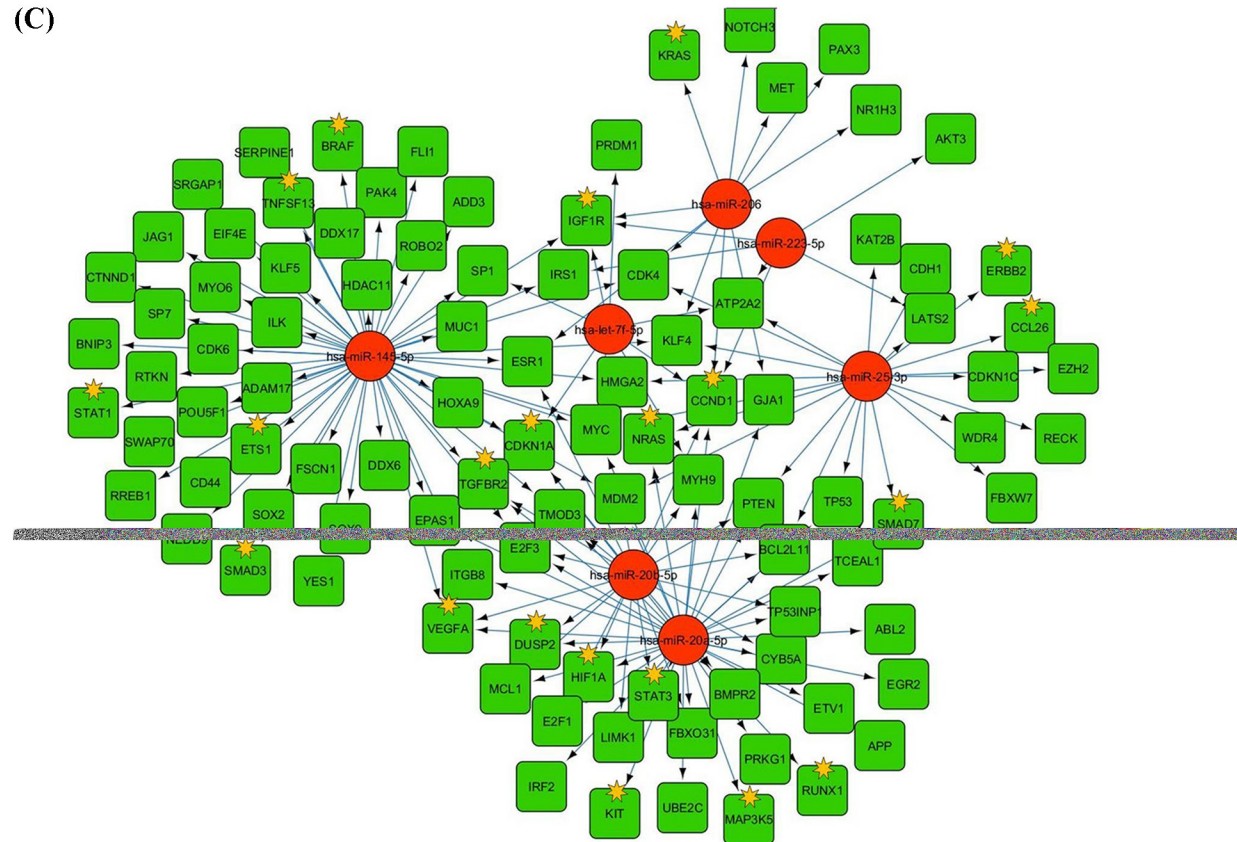

**Fig 2.** Significantly enriched GO term (A) and KEGG pathway (B) of seven DE miRNAs. (C) Visualization of miRNAs and their associated target genes with Cytoscape. Red: miRNAs; Green: mRNAs; and yellow star: cytokine-associated genes.

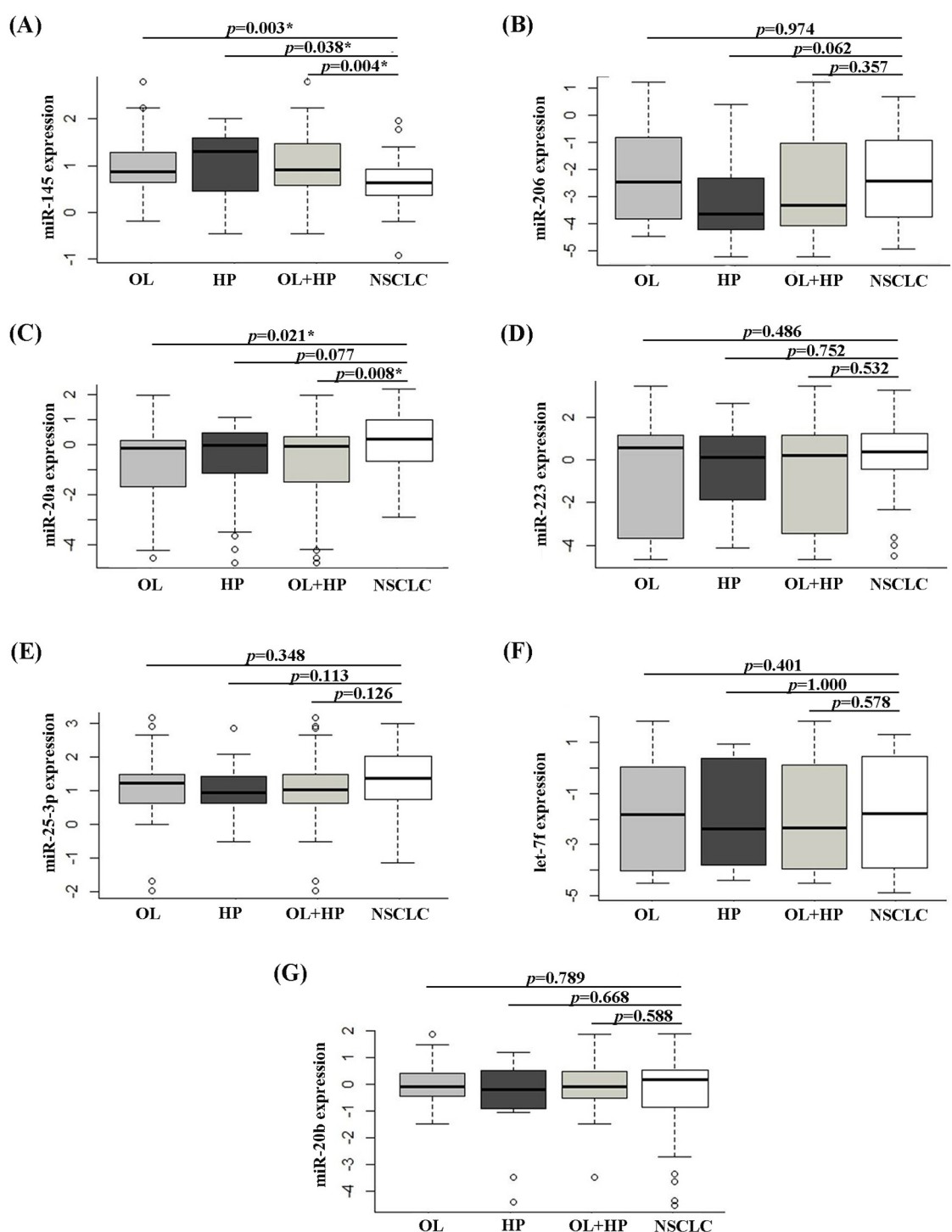

**Fig 3. Relative expression of miR-145 (A), miR-206 (B), miR-20a (C), miR-223 (D), miR-25-3p (E), let-7f (F), and miR-20b (G) using qRT-PCR in the serum of NSCLC patients compared with OL patients, HP, and all controls (OL + HP).** (*) indicates significant difference of $p < 0.05$.

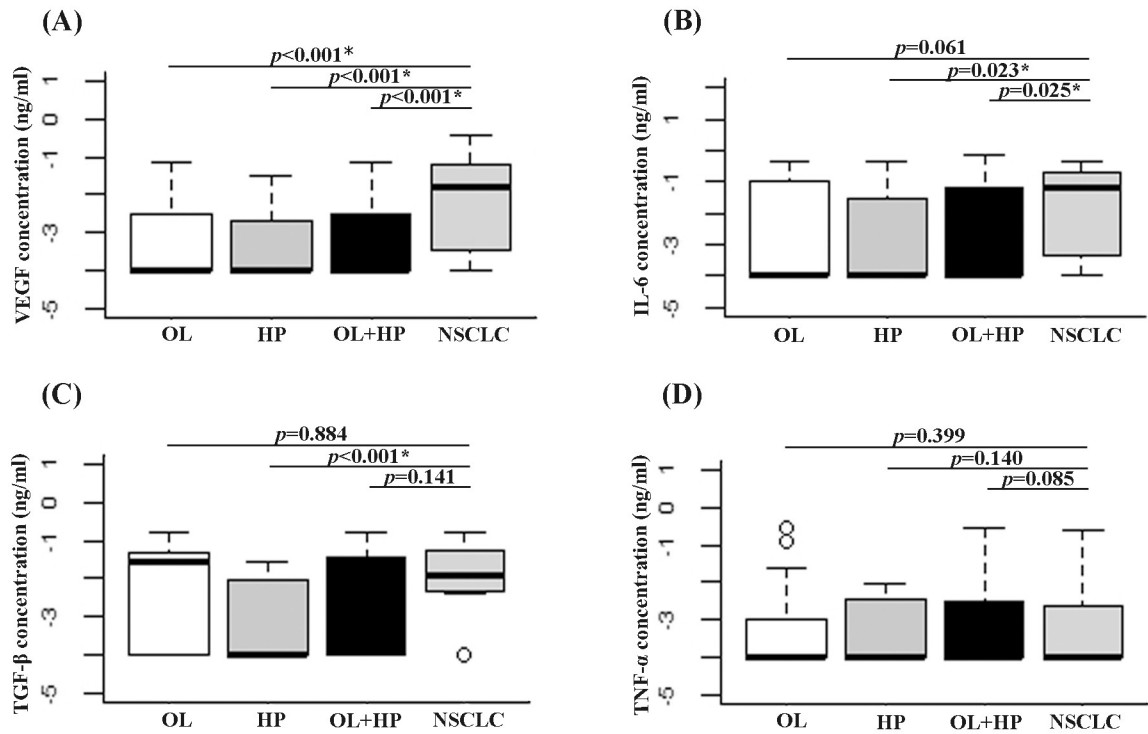

**Fig 4. The concentration of VEGF (A), IL-6 (B), TGF-β (C), and TNF-α (D) using ELISA in the serum of NSCLC patients compared with OL patients, HP, and all controls (OL + HP).** (*) indicates a significant difference of $p < 0.05$.

and HP ($p < 0.001$) (Fig 4A). A significantly elevated concentration was also found in NSCLC patients compared with the HP group for IL-6 ($p = 0.023$) and TGF-β ($p < 0.001$), and all controls for IL-6 ($p = 0.025$) (Fig 4B and 4C). There was no significant difference in serum TNF-α concentration between NSCLC patients and controls (Fig 4D).

## Correlation between miRNA expression and cytokine level

We evaluated the correlation between miRNA expression and cytokine concentration in the serum of NSCLC patients by Spearman's correlation. A moderate correlation was shown between the concentration of TGF-β and the expression level of miR-20a (r = −0.537; $p = 0.002$) and miR-223 (r = 0.428; $p = 0.015$), in both negative and positive directions, respectively (Fig 5A and 5B). We also detected a weak, positive correlation of VEGF concentration with miR-20a expression (r = 0.376; $p = 0.037$) (Fig 5C), whereas miR-223 expression was weakly, negatively correlated with serum VEGF concentration (r = −0.355; $p = 0.046$) (Fig 5D). However, the other miRNAs did not exhibit any correlation with cytokines (Table 3).

## Discussion

In the present study, the lists of DE miRNAs were identified and then validated in clinical samples. Among 16 lung cancer miRNA profiling studies, eight miRNAs were consistently reported to be differentially expressed in at least three studies. In contrast, seven miRNAs were reported to be differentially expressed with inconsistent direction (either up- or down-regulation) in at least four studies. In the validation study, miR-145 and miR-20a showed a significant difference between NSCLC patients and controls. We also found that the cytokine concentrations of VEGF, IL-6, and TGF-β, but not TNF-α, were higher in NSCLC patients

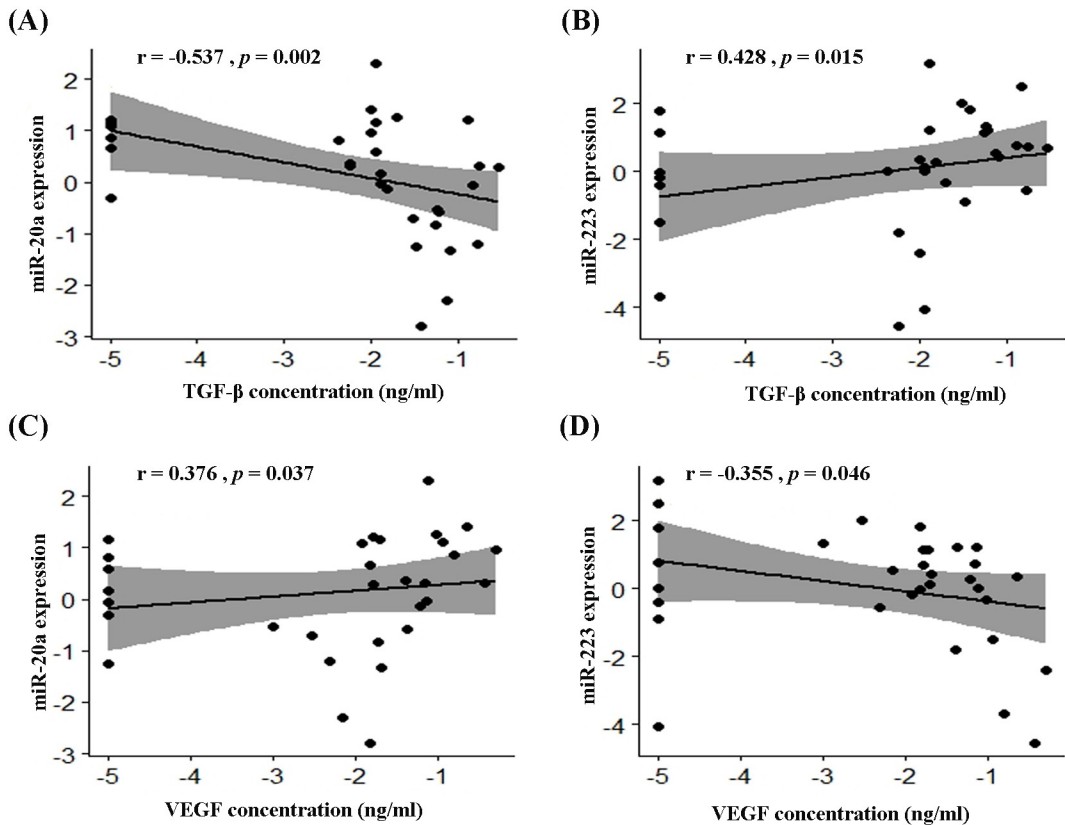

**Fig 5. Scatter plots with regression line and 95% CI (gray shade) between miRNA expression and cytokine level in the serum of patients with NSCLC.** (A) miR-20a versus TGF-β, (B) miR-223 versus TGF-β, (C) miR-20a versus VEGF, and (D) miR-223 versus VEGF. (*) indicates significant difference of $p < 0.05$ from Spearman's rank correlation.

than in the control groups. In addition, the expression of miR-20a and miR-223 were correlated with the concentrations of TGF-β and VEGF in NSCLC patients.

Although several miRNA profiling studies to identify potential biomarkers for lung cancer have been conducted, the lists of DE miRNAs are not consistent. To deal with this issue, identification based on the vote-counting strategy, which has not yet been reported for lung cancer, was used. We demonstrated that the number of consistent DE miRNAs (79%, 181/229

**Table 3. Correlation analysis between miRNA expression and cytokine concentration in the serum of NSCLC patients.**

| miRNA | IL-6 | | VEGF | | TGF-β | | TNF-α | |
|---|---|---|---|---|---|---|---|---|
| | r | p-value | r | p-value | r | p-value | r | p-value |
| miR-145 | -0.062 | 0.738 | -0.301 | 0.107 | -0.023 | 0.899 | -0.183 | 0.467 |
| miR-206 | -0.008 | 0.967 | -0.075 | 0.682 | -0.084 | 0.648 | -0.114 | 0.653 |
| miR-20a | -0.062 | 0.738 | **0.376** | **0.037***  | **-0.537** | **0.002***  | -0.245 | 0.327 |
| miR-223 | 0.037 | 0.841 | **-0.355** | **0.046***  | **0.428** | **0.015***  | 0.268 | 0.282 |
| miR-25-3p | 0.003 | 0.988 | 0.237 | 0.191 | -0.272 | 0.132 | -0.110 | 0.665 |
| let-7f-5p | 0.095 | 0.606 | 0.147 | 0.423 | -0.022 | 0.904 | -0.186 | 0.459 |
| miR-20b | -0.156 | 0.410 | 0.085 | 0.645 | -0.011 | 0.951 | 0.076 | 0.766 |

IL-6, interleukin-6; VEGF, vascular endothelial growth factor; TGF-β, transforming growth factor β; TNF-α, tumor necrosis factor-α.

(*) indicates significant difference of $p < 0.05$.

miRNAs) was higher than those of inconsistent DE miRNAs (21%, 48/229 miRNAs). This result indicated a high degree of concordance among the studies. Inconsistent DE miRNAs could potentially be explained by several factors, such as the diversity of specimens, histological subtypes, clinical stages, and profiling platforms.

From the identification, miR-145 was consistently found to be up-regulated in four studies [26, 27, 29, 34]. However, our validation results in the clinical samples demonstrated the down-regulation of miR-145 in NSCLC patients compared with controls. Previous studies have shown that overexpression of miR-145 increased cell death of prostate cancer cells [39], and inhibited the invasion and metastasis of osteosarcoma cells [40]. Moreover, low expression levels of serum miR-145 predicted poor survival in patients with ovarian cancer [41]. In lung cancer, a decreased expression of miR-145 in tissues was found to have a significantly worse prognosis [42] and associated with a shorter time to relapse in NSCLC [43]. Our finding in clinical validation was consistent with these previous studies, indicating that the miR-145 might play a functional role as tumor-suppressive miRNA.

MiR-20a was found to be up-regulated in four studies [23, 27, 29, 34] and down-regulated in two studies [26, 32]. Our validation study supported the evidence for the up-regulation of miR-20a in NSCLC compared with controls. Although several studies have revealed the aberrant expression of miR-20a in various types of cancer, data on its function is inconsistent. For example, miR-20a overexpression inhibited the invasion of endometrial cancer cells [44] and the proliferation of neuroblastoma cells [45]. In contrast, miR-20a has been found to promote proliferation, invasion, and metastasis in hepatocellular carcinoma [46] and colorectal cancer cells [47]. Regarding clinical studies, increased expression of plasma/serum miR-20a was found in glioblastoma [48], gastric cancer [49], and NSCLC [50]. It was also associated with overall patient survival and disease-free survival. In addition, our bioinformatics analysis revealed that miR-20a regulated several cancer-related genes, particularly tumor suppressor genes, such as PTEN, BCL2L11, FBXO31, and DUSP2. This evidence was in accordance with our result in clinical samples that indicated an oncogenic property of miR-20a in lung cancer.

One important discrepancy between the results in our identification and clinical validation is the stage distribution of NSCLC patients. The studies of Gao *et al.* (2016) [24], Geng *et al.* (2014) [29], and Foss *et al.* (2011) [35] include early-stage NSCLC, while most of the patients in our study were in an advanced stage. This explanation was supported by the study of Aiso *et al.* [51], demonstrating that the serum expressions of miR-145 and miR-20a were higher in the early stages than the advanced stages of NSCLC.

Cytokines have been proposed as blood-based biomarkers and therapeutic targets, because they play essential roles in cancer development, including lung cancer [5]. The increased serum concentration of several cytokines, including VEGF, IL-6, TGF-β, and TNF-α, have been proposed as lung cancer biomarkers [5]. In the present study, we confirmed the increased serum concentration of all cytokines, except TNF-α in NSCLC compared with HP. Since the overestimation of the test could be found if only healthy persons are used as controls [52], both HP and patients with OL patients who may have symptoms similar to lung cancers were also included in this study for comparison.

There are accumulating studies showing that miRNAs associate with tumorigenesis through the regulation of cytokine production [11–13]. We found a moderate correlation between TGF-β concentration and expression levels of miR-20a and miR-223, and a weak correlation between VEGF concentration and expression levels of miR-20a and miR-223. Several studies have demonstrated that miR-20a regulates related genes in TGF-β-signaling, such as TGIF2, E2F5, MYC, ALK5, and TGFBR2 [11, 53]. In addition, some evidence has revealed that miR-20a is associated with VEGF production in breast cancer [12]. For miR-223, Berenstein *et al.* (2016) [10] have shown that decreased expression of miR-223 increases VEGF

expression in bone marrow mesenchymal stromal cells. Liu *et al.* (2018) [54] have observed that miR-223 is increased in TGF-β1-stimulated cardiac fibroblasts. Our current results support the findings of theses previous studies, showing that miR-20a and miR-223 may regulate genes associated with TGF-β and VEGF.

There are some limitations to our present study. First, we included advanced-stage NSCLC for clinical validation. Since the DE miRNAs were identified from miRNA profiling conducted from various stages of NSCLC, the samples from patients with different stages should be more suitable than those from advanced stages of disease in our study. Second, the number of clinical samples, both NSCLC, and controls, is small. This low number may be subject to a type II error (false negative). Therefore, a large clinical sample should be used to assess the diagnostic performance of miR-145 and miR-20a for clinical utility.

## Conclusions

We identified miR-145 and miR-20a as potential biomarkers for NSCLC. Our study provides supporting evidence of increased tumor-promoting cytokines in NSCLC, which are correlated with miR-20a and miR-223. Our results suggest that the regulation of tumor-promoting cytokines, through miR-20a and miR-223, might be a new therapeutic approach for lung cancer.

## Supporting information

**S1 Table. List of the genes associated with the consistently up-regulated miRNA.**
(PDF)

**S2 Table. List of the genes associated with the consistently down-regulated miRNA.**
(PDF)

**S3 Table. List of the genes associated with the inconsistency reported miRNA.**
(PDF)

## Acknowledgments

Research center for cancer control in Thailand are acknowledged for their research facilities. We thank the office of International Affairs, Faculty of Medicine, Prince of Songkla University for English language editing.

## Author Contributions

**Conceptualization:** Sarayut Lucien Geater, Paramee Thongsuksai, Pritsana Raungrut.

**Data curation:** Keson Trakunran, Sarayut Lucien Geater, Warangkana Keeratichananont, Pritsana Raungrut.

**Formal analysis:** Pichitpon Chaniad, Keson Trakunran, Paramee Thongsuksai, Pritsana Raungrut.

**Investigation:** Pichitpon Chaniad, Pritsana Raungrut.

**Supervision:** Pritsana Raungrut.

**Validation:** Pichitpon Chaniad.

**Writing – original draft:** Pichitpon Chaniad, Pritsana Raungrut.

**Writing – review & editing:** Paramee Thongsuksai, Pritsana Raungrut.

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
