## [Decision Letter · Decision Letter 0]

7 Aug 2020

PONE-D-20-07179

Serum miRNAs associated with tumor-promoting cytokines in non-small cell lung cancer

PLOS ONE

Dear Dr. Raungrut,

Thank you for submitting your manuscript to PLOS ONE. After careful consideration, we feel that it has merit but does not fully meet PLOS ONE’s publication criteria as it currently stands. Therefore, we invite you to submit a revised version of the manuscript that addresses the points raised during the review process.

We look forward to receiving your revised manuscript.

Kind regards,

Esra Bozgeyik, PhD

Academic Editor

PLOS ONE

Journal Requirements:

Review Comments to the Author

Reviewer #1: Although the study is well-designed, the manuscript and figures are clear and the results look consistent, the biggest handicap of this work is an extremely limited clinical sample size. The authors will have to tune down very much.

Additional comments are as follows:

Q1. Page 2 line 5 authors state that: NSCLC abbreviation is not clear for the reader. In an abstract should be writing at complete form "non-small cell lung cancers (NSCLC)".

Q2. Page 6 Line 102: "had" is repeated.

Q3.In the Materials and Methods Section, such information should be added to describe the control group. The authors state that using A p-value in 0.05 but on page 12 using for serum cytokine concentration P-value in 0.001. Please clarify what value was considered for statistical significance?

Q4 How do you inhibit cytokines? I think it is very important to add some sentences to answer this question.

Reviewer #2: The paper is technically sound and well-detailed.

Concerning the statistical analyses, my knowledge about this domain is quite limited as I am less familiar with it.

The limitations of the study have also been well-defined.

---

## [Author Response · Author response to Decision Letter 0]

2 Oct 2020

Responses to the reviewers’ comments

Reviewer #1: 

Although the study is well-designed, the manuscript and figures are clear and the results look consistent, the biggest handicap of this work is an extremely limited clinical sample size. The authors will have to tune down very much. 

Answer:

We are very concerned about this point. The limitation regarding small sample size has already been mentioned in a topic of discussion in the page 18, line 320-323 as following:

“Second, the number of clinical samples, both NSCLC, and controls, is small. This low number may be subject to a type II error (false negative). Therefore, a large clinical sample should be used to assess the diagnostic performance of miR-145 and miR-20a for clinical utility.”

Additional comments are as follows:

Q1. Page 2 line 5 authors state that: NSCLC abbreviation is not clear for the reader. In an abstract should be writing at complete form "non-small cell lung cancers (NSCLC)".

Answer:

We have put a complete form of NSCLC as shown in an abstract the page 2, line 4-5.

Q2. Page 6 Line 102: "had" is repeated.

Answer:

We have deleted a repeated word as shown in a subtopic of study subjects and sample collection in the page 6, line 101.

Q3. In the Materials and Methods Section, such information should be added to describe the control group. 

Answer:

All information of the control group both other lung disease (OL) and healthy person (HP) groups had already been stated in a subtopic of study subjects and sample collection in Page 6 line 98-102 as following:

“For the control groups, subjects who were age- and gender-matched with the NSCLC group were included. The OL group were patients who had symptoms similar to lung cancer, such as tuberculosis, bronchiectasis, interstitial lung disease, pneumonia, and chronic obstructive pulmonary disease. The HP group was recruited from people who had an annual checkup for two consecutive years.”

Q4. The authors state that using A p-value in 0.05 but on page 12 using for serum cytokine concentration P-value in 0.001. Please clarify what value was considered for statistical significance?

Answer:

We used a p-value less than 0.05 for statistical significance as we have mentioned in a subtopic of statistical analysis in the page 7-8, line 137-138 as following:

“A p-value of less than 0.05 was considered statistically significant.”

Q5. How do you inhibit cytokines? I think it is very important to add some sentences to answer this question.

Answer: 

Our results showed significant correlation between miR-20a and miR-223 with VEGF and TGF-β (Table 2). There are also several previous studies demonstrated the functional relationships of these two miRNAs and the cytokines. We have already discussed this issue by addressing such evidences in a topic of discussion in the page 17-18, line 305-316 as following:

“There are accumulating studies showing that miRNAs associate with tumorigenesis through the regulation of cytokine production [11-13]. We found a moderate correlation between TGF-β concentration and expression levels of miR-20a and miR-223, and a weak correlation between VEGF concentration and expression levels of miR-20a and miR-223. Several studies have demonstrated that miR-20a regulates related genes in TGF-β-signaling, such as TGIF2, E2F5, MYC, ALK5, and TGFBR2 [11, 53]. In addition, some evidence has revealed that miR-20a is associated with VEGF production in breast cancer [12]. For miR-223, Berenstein et al. (2016) have shown that decreased expression of miR-223 increases VEGF expression in bone marrow mesenchymal stromal cells [10]. Liu et al. (2018) have observed that miR-223 is increased in TGF-β1-stimulated cardiac fibroblasts [54]. Our current results support the findings of theses previous studies, showing that miR-20a and miR-223 may regulate genes associated with TGF-β and VEGF.”

Reviewer #2: 

The paper is technically sound and well-detailed. Concerning the statistical analyses, my knowledge about this domain is quite limited as I am less familiar with it. The limitations of the study have also been well-defined.

Answer: 

Thanks for your kind words and valuable suggestion.

Additional comments:

Q1. English should be revised 

Answer: 

We have edited English writing of the entire manuscript by using the professional editing and proofreading service. 

Q2. Abbreviations such as DE (differentially expressed) should be specified

Answer:

It has been specified as shown in the page 4, line 49 and ‘Revised manuscript with track change’. 

Q3. In the section methods, Quantitative real-time PCR (qRT-PCR) should be more detailed, and sequences of primers used to amplify miRNA should be provided in a supplementary table.

Answer:

We could not get sequences of miRNA primers because they are commercial miRNA-forward primers from Qiagen. Regarding the condition of qRT-PCR, we have put more detail in a subtopic of Quantitative real-time PCR (qRT-PCR) in the Page 6-7 line 109-123 as following:

“The total RNA from serum was extracted using Trizol® LS (Invitrogen, California, USA) according to the manufacturer’s protocol. The quantity and quality of the total RNA were measured using a NanoDrop� ND-1000 UV-Vis spectrophotometer (Thermo Scientific, Massachusetts, USA) at an optical density ratio of A260/280 nm and A260/230 nm. The total RNA (50 ng) was reverse transcribed into complementary DNA (cDNA) by the miScript II RT kit (Qiagen, Hilden, Germany). Reverse transcription was performed by Thermal cycler (Bio-Rad, California, USA) at 37 °C for 60 mins, and 95 °C for 5 mins. The cDNA was diluted by adding 200 μl of RNase-free water. Subsequently, miRNA amplification was performed using a Bio-Rad CFX96 qPCR system (Bio-Rad, California, USA) with the miScript SYBR® green PCR kit (Qiagen, Hilden, Germany) according to the manufacturer’s protocol. Primers for miR-145-5p, miR-206-5p, miR-20a-5p, miR-223-5p, miR-25-3p, let-7f-5p, miR-20b-5p, and U6 small nuclear RNA (RNU6), were used as internal controls, and obtained from Qiagen (Qiagen, Hilden, Germany). Data were presented as the cycle threshold (Ct) value, which was determined using the default threshold settings. The difference between the Ct value (ΔCt) of the miRNA of interest and RNU6 was calculated, and the relative expression was presented using the 2−∆CT method.”

Q4. Results of miRNA expression and cytokine concentration obtained by qRT-PCR and ELISA, respectively, and used for statistical analyses were not available.

Answer:

We have already mentioned about statistical analyses for miRNA expression and cytokine concentration in a subtopic of statistical analysis in the page 7 line 132-136 as following:

“Data distribution of relative miRNA expression and cytokine concentration was assessed by the Shapiro–Wilk normality test. Outliers of the data were detected using the Grubbs’ test. Differences in the data between the groups of subjects were analyzed by the Student’s t-test if the data had a normal distribution, and by the Wilcoxon–Mann–Whitney test in cases of non-normal distribution.”

Q5. In table 2, references of studies should be added.

Answer:

References of the studies have been added in the table 2 as shown in the page 11. 

Q6. Discussion should be revised in term of sentences and use of English.

Answer:

We have edited English writing of the entire manuscript by using the professional editing and proofreading service.

---

## [Decision Letter · Decision Letter 1]

19 Oct 2020

Serum miRNAs associated with tumor-promoting cytokines in non-small cell lung cancer

PONE-D-20-07179R1

Dear Dr. Raungrut,

We’re pleased to inform you that your manuscript has been judged scientifically suitable for publication and will be formally accepted for publication once it meets all outstanding technical requirements.

Kind regards,

Esra Bozgeyik, PhD

Academic Editor

PLOS ONE

---

## [Editor Report · Acceptance letter]

21 Oct 2020

PONE-D-20-07179R1 

Serum miRNAs associated with tumor-promoting cytokines in non-small cell lung cancer

Dear Dr. Raungrut:

I'm pleased to inform you that your manuscript has been deemed suitable for publication in PLOS ONE. Congratulations! Your manuscript is now with our production department. 

Kind regards, 

on behalf of

Dr. Esra Bozgeyik 

Academic Editor

PLOS ONE